# Ecological, Cultural, and Geographical Implications of *Brahea dulcis* (Kunth) Mart. Insights for Sustainable Management in Mexico

**Cloe X. Pérez-Valladares** [1], **Ana I. Moreno-Calles** [2,*], **Alejandro Casas** [3],
**Selene Rangel-Landa** [3], **José Blancas** [4], **Javier Caballero** [5] and **Alejandro Velazquez** [1]

1  Centro de Investigaciones en Geografía Ambiental (CIGA), Universidad Nacional Autónoma de México, Antigua Carretera a Pátzcuaro 8701, Morelia 58190, Michoacán, Mexico; cvalladares@pmip.unam.mx (C.X.P.-V.); alex@ciga.unam.mx (A.V.)
2  Escuela Nacional de Estudios Superiores (ENES), Universidad Nacional Autónoma de México, Antigua Carretera a Pátzcuaro 8701, Morelia 58190, Michoacán, Mexico
3  Instituto de Investigaciones en Ecosistemas y Sustentabilidad (IIES), Universidad Nacional Autónoma de México, Antigua Carretera a Pátzcuaro 8701, Morelia 58190, Michoacán, Mexico; acasas@cieco.unam.mx (A.C.); srangel@cieco.unam.mx (S.R.-L.)
4  Centro de Investigación en Biodiversidad y Conservación (CIByC), Universidad Autónoma del Estado de Morelos, Av. Universidad 1001, Colonia Chamilpa, Cuernavaca 62209, Morelos, Mexico; jose.blancas@uaem.mx
5  Instituto de Biología, Universidad Nacional Autónoma de México, Cto. Zona Deportiva, s/n, Ciudad Universitaria, Ciudad de Mexico 04510, Mexico; jcaballero@ib.unam.mx
*  Correspondence: isabel_moreno@enesmorelia.unam.mx

**Abstract:** Palm plants provide important benefits for rural communities around the world. Of the 95 native palm species in Mexico, *Brahea dulcis* (*Soyate* palm) has been tagged as an important resource for many Mesoamerican ethnical groups. Scientific and empirical knowledge concerning *Soyate* is thematically fragmented and disassociated, meaning that sound sustainable management is far from established. Research of over 20 years has permitted us to document ecological, cultural and geographical outcomes of *B. dulcis;* thus, the present paper aims at compiling all knowledge on *Soyate* to eventually guide its long-term management. It was conducted in two stages: firstly, it comprised a thorough review of previous studies on the management of *B. dulcis* in Mexico; secondly, we integrated unpublished outcomes obtained from fieldwork, including participatory ground-truth validation and semi-structured interviews obtained from local ethnic groups. Five factors guided our compilation effort: (i) biological and ecological information, (ii) cultural importance, (iii) economic triggers, (iv) traditional management, and (v) ecological and ecogeographical implications of *Soyate* palm management. The present paper confirms that *B. dulcis* is an important cultural resource whose utilization can be traced back over 10,000 years. The leaves of *Soyate* are the most useful part of the palm and were profusely used in the past for thatching roofs and weaving domestic and agricultural objects. Currently, however, palm-leaf weaving is primarily oriented toward satisfying economic needs. We depicted ten management practices aimed at favoring palm availability. Most of these management practices have enhanced sustainable palm leaf harvesting; however, these practices harbor spatial trends that turn highly diverse habitats into *Soyate*-dominated spaces. To conclude, we propose a framework to describe sound and sustainable *Soyate* management in the light of the current long-term *Soyate*–human relationship. It is here acknowledged that *Soyate* has played and continues to play a critical socioeconomic and cultural role for many ethnical groups in Central Mexico. Nonetheless, emerging challenges concerning the sustainability of the whole socioecological system at a landscape level are yet to be overcome.

**Keywords:** non-timber forest products; traditional palm management; palm ethnobotany; landscape domestication; fiber handcrafts; ecological inheritance

## 1. Introduction

Non-modern practices performed on individuals, populations, and ecosystems directed toward the use, maintenance or increase in the availability of resources are forms of traditional management [1–3]. Such actions are performed at different spatial and temporal scales and operate at different levels of social organization [4]. These practices and strategies are constructed from perceptions and knowledge of nature, as well as complex socio-historical processes influenced by the human–nature relationships of cultures [5]—a knowledge that is gradually improving and constantly built from experimenting in specific places [6,7]. Traditional management has the purpose of establishing appropriate conditions for obtaining natural resources or environmental benefits to satisfy human needs [1–4]. Five general types of management practices are commonly recognized, namely, gathering (which may include the selection of phenotypes, as well as the regulations or social organization required to carry this out), tolerance, promotion, protection, and cultivation [1,2]. These five actions may be performed in situ, in natural areas where the species occur, or may be extracted from the original distribution sites and manipulated ex situ, in areas such as home gardens and other agroforestry systems [4]. Although such practices are commonly directed to specific plant resources, with time, the continual practice of these actions results in progressive modifications of the environment that improve or reduce resource availability. Thus, management has ecological, cultural, and spatial dimensions [4,8,9].

Examples of the effect of traditional managing palms are the atypical presence and abundance of some species of *Astrocaryum*, *Acrocomia*, *Attalea* and *Bactris* related with ancient human settlements in Amazonia [10,11]. These species are used in searching for archaeological remains since they are considered indicators of ancient human settlements. The continuous selection of individuals with desirable characteristics has given way to the domestication of some species populations and to the consequent differentiation of varieties responding to different human needs. This has been the case for date palm (*Phoenix dactylifera*) [12], coconut palm (*Cocos nucifera*) [13], and pejibaye palm (*Bactris gasipaes*) [14]. The traditional management of palms as non-timber forest products (NTFPs) has been pointed out as a strategy that could allow ecosystem conservation, while generating economic benefits for communities [15–17]. For this reason, scientific surveys aimed at assessing management implications should holistically analyze the entire geographical context, addressing spatial, ecological, and cultural aspects. Such a landscape perspective has been poorly addressed in the global study of NTFPs.

Traditional palm management is performed by rural communities in tropical regions around the world [18,19] where these resources fulfill several necessities: they are a source of food, fiber, construction material and oil, as well as raw materials for handcrafts. This has made the Arecaceae family the third most important cultural and economic botanical group worldwide, after only grasses and legumes [19]. Records of human groups using palms in the Americas go back almost to 12,000 years before present (BP) [10]. Due to their remarkable ecological and cultural importance, Balick [11] considered palms to be as important as maize in the development of precolonial societies in the Americas and suggested that their prehistoric use is deeply related to the dispersion, selection, domestication, and extinction of species, which has important implications for explaining their current geographical distribution. Traditional palm management frequently occurs in arid and sub-humid ecosystems, where the selling of palm-derived products sometimes represents the only source of income; thus, this is regarded as critical for monetary compensation [11,20]. Furthermore, palms have been present in a diversity of aspects of the cultural life of indigenous, afrodescendant and mestizo communities in Latin America [10,11,21–24], playing an important role in their identities due to the large amount of resources they provide [10,11,23,25].

Mexico has not been an exception regarding the use of palms since ancient times. The earliest record of palms being used by humans in Mexico dates back nearly 11,950 years BP [10], and a large amount of archeological remains has been identified in caves and other sites [26]. These records suggest that palms were originally used for the confection of fiber strips [27]. The cultural importance of palms among Mexican people is reflected in the inclusion of these plants in daily life and ritual–ceremonial practices [3], as well as in the traditional knowledge of communities regarding their uses [28,29]. Several indigenous groups are recognized through their bond with palms: among the Maya people, the use of guano palm (*Sabal* spp.) and its leaves has been practiced since ancient times, mainly for thatching traditional houses [29]. The *Ñuu savi* (Mixtec), *Xwja* (Ixcatec), and *Ngiwa* (Popolocan) ethnic groups have been dubbed "the eternal palm weavers" due to the cultural importance of weaving hats, "petates" (a kind of palm woven mat), and "tenates" (a palm woven container), as well as other diverse handcrafts using *Brahea dulcis* palm leaves [30–34]. In addition, there are several cultural groups which are deeply bound with this palm, such as the *Nahua*, *Nduudu yu* (Cuicatec), *Binni zá a* (Zapotec), *Ayuukjä'äy* (Mixe), *Runixa ngiigua* (Chocho), *Ha shuta enima* (Mazatec), and *Hñä hñü* (Otomi). Thus, palm management has been culturally connected with the identity of the Mesoamerican people, through this acquiring its traditional representation [35].

Similarly, as in other countries of South America [23], palms in Mexico have mostly been used as material for construction, food, and the production of tools and utensils of domestic and agricultural use. The palm's leaves, fruits, and apical meristems are the most used parts [21,29,36–40]. The plant fiber which makes up the leaves is of primary importance in roof thatching and the production of household goods [29,41]. According to Belcher [16], plant fibers are the most important NTFPs, after medicinal and food plants. In Mexico, the *Brahea* and *Sabal* species stand out for their value as plant fiber sources [21,28,29,36–41]. *Brahea* is of primary importance since it is a predominantly Mexican endemic genus, primarily occurring in dry climates and semiarid environments [42]. In this context, *Brahea* plays an important role as a strategic resource in rural household economies [41]. Prior to this study, several local surveys documented the economical relevance of *B. dulcis* (*Soyate*) for rural communities [20,21,37,41,43,44]. The outreach of these surveys, however, did not permit the provision of a national perspective in order to assess implications for sustainability.

The present paper aims at compiling ecological, cultural, and geographical information relating to the traditional management of *Brahea dulcis* to feature insights regarding its sustainability. We argue that the *Soyate* palm socio-ecological system might be regarded as one of the most culturally outstanding examples in Mesoamerica. We further discuss *Brahea dulcis* as a model for exploring the implications of in situ traditional palm management for reshaping the region at a landscape level.

## 2. Methods

Our research was conducted in two phases: firstly, a bibliographic search was conducted from February to August 2018 using the Scopus, Web of Sciences, and Google Scholar databases using "*Brahea dulcis* México", "palm management Mexico", "palm traditional management", "manejo tradicional *Brahea dulcis*", and "manejo palmas *Mexico*" as keywords. We additionally performed a continuous review of new sources based on the literature referred to in the documents obtained from these databases. These results are presented in Supplementary File S1.

We secondly integrated unpublished outcomes obtained from fieldwork, including participatory ground-truth validation and semi-structured interviews obtained from Guerrero, Puebla, and Oaxaca local ethnical groups. Five factors guided our compilation effort: (i) general biological and ecological information; (ii) cultural importance; (iii) economic triggers; iv) traditional management; and (v) landscape implications of the *Soyate* palm management. By means of participative observation [45] and open and semi-structured interviews [46] held with weavers, harvesters, sellers, and local authorities, we documented detailed aspects of the management of *Brahea dulcis*. As part of the dialogues established with stakeholders, a technical report on palm stand management was also provided by the administration of the Tehuacán-Cuicatlán Biosphere Reserve.

An update was made of the palm species list reported for Mexico by Quero (1994) [38], and we reviewed the inventory and general information of the Mexican taxa of the Arecaceae family in order to be able to refer to this study within the context of palm cultural diversity in Mexico (Supplementary File S2).

A map of the known distribution of *Brahea dulcis* was created in a Geographical Information System (the open source Quantum Geographical Information System, QGIS www.qgis.org/es/site/; an official project of Open Source Geospatial Foundation) based on georeferenced records obtained from the Global International Biodiversity Facility (https://www.gbif.org/). The results of the bibliographic search were used to complement the map of *B. dulcis* studies in Mexico. On this map, the geographic location of the studies is indicated by the ID assigned in the table of bibliographic results (Supplementary File S1).

Integrated outcomes were further discussed in light of their implications for the sustainable management of *Soyate* and geographic implications at the landscape level.

## 3. Results

### 3.1. Brahea dulcis (Soyate Palm). Biological and Ecological Information

A total of 95 palm species belonging to 21 genera are listed for Mexico; of these, *Brahea* stands out as a predominantly Mexican genus. The *Brahea dulcis* species is that with the highest number of records of use among ethnic groups in Mexico (Supplementary File S2). Although the known distribution range of *B. dulcis* is from northern Mexico to Central America (Figure 1A), all results from the bibliographic search are studies conducted in Central–Southern Mexico (Figure 1B).

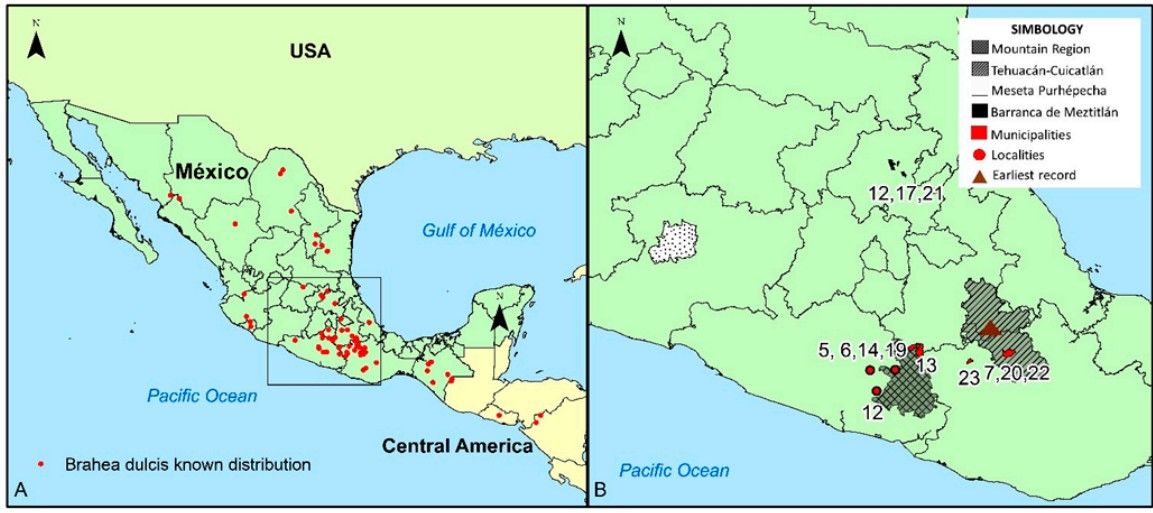

**Figure 1.** (**A**) Known *Brahea dulcis* distribution. (**B**) The geographic location of studies relating to *Brahea dulcis* management in Mexico. Numbers indicate the ID of studies referred to in Table 2.

*Brahea* is one of the least-known palm genera in the Americas [19], even though *Brahea dulcis* is the most abundant and widespread palm species in Mexico [43]. This species is mainly found in limestone soils, predominantly at elevations between 800 and 1600 m. It occurs in tropical dry forests, oak forests, and xerophytic scrubland from sub-humid to semidesert climates [20,28,38,47]. *Brahea dulcis* constitutes vegetation patches known as palm stands or *palmares*. According to several authors, *palmares* derive from management, through which the dominance of *B. dulcis* is favored by the selective removal of competing standing vegetation and man-made fires [20,37]. These management practices may increase palm abundance by up to 12-fold [48].

The phenotypic plasticity of *Soyate* palm has been reported [28,37]. Differences in the size of stems and leaves may depend on the developed life habits, which can be solitary or colonial (Figure 2).

This variable phenotype is malleable through management, where arboreal palm stands develop from solitary life habits with tall individuals or *Soyacahuite* (term derived from the Náhuatl meaning *soyatl* = palm and *cuahuitl* = tree; or, in other words, arboreal palms). Thus, arboreal palm-dominated stands are locally known as *Soyacahuiteras*, while the stands of shrubby colonial palms are called as *Manchoneras* (their nature and relation to management are discussed below).

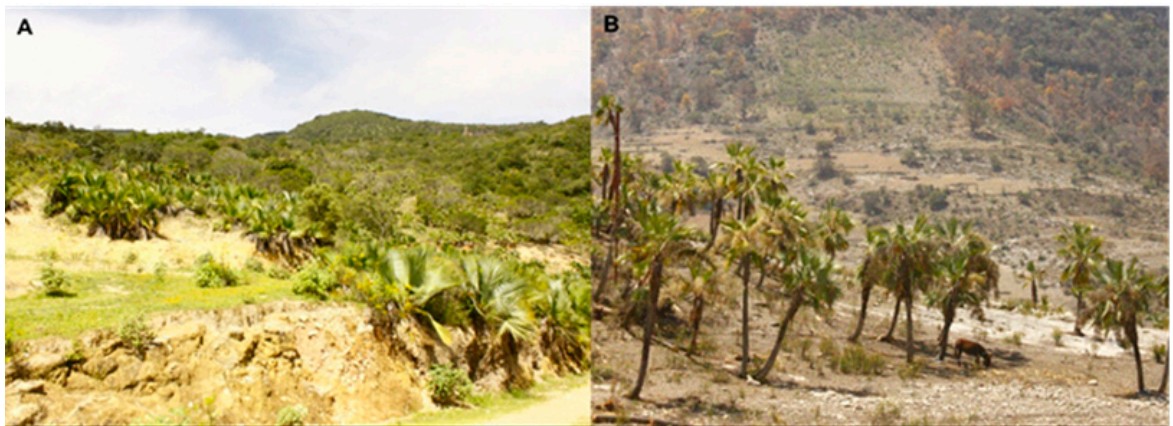

**Figure 2.** Morphotypes of *Brahea dulcis*. (**A**) Low-stature colonial life habit. Palm stands with a dominance of this phenotype are named *Manchoneras.* (**B**) High-stature individual life habit. Palm stands dominated by this phenotype are named *Soyacahuiteras.* Photo credit: Cloe Xochitl Pérez-Valladares.

*3.2. Cultural Relevance*

The earliest archaeological record of *Brahea dulcis* associated with human use dates back ca. 11,950 years BP in the caves of the Tehuacán Valley [26]. The discovered remains correspond to fragments of palm leaves that are assumed to have served as a fiber source since antiquity and were used to produce fiber strips by the earliest hunter–gatherer human groups in Central–Southern Mexico [26,27]. During pre-Hispanic times, Mesoamerican ethnical groups paid tribute to the Aztec Empire with objects made of *B. dulcis*. Products such as *tenates* and *petates* were used as tribute; the former were and still are useful and common containers, whereas the latter were regarded as valuable objects in wars, protecting armies from the sun during the day and used for rest at night [49,50]. The exchange of palm products for gold and other goods was documented in the 16th century and was carried out long before [49]. Dominican friars introduced the practice of hat weaving—a production that became an important industry during the 19th century and most of the 20th century [51,52].

Currently, *B. dulcis* has a wide variety of uses. Firstly, indirect food sources such as edible insects hosted by the palm are used as protein sources, such as for *soyacuilin* and *sochiahuatl*, insects from the genus *Rhynchophorus* (order Coleoptera) whose larvae are consumed after being cooked [28]. All the main parts of the palm are used, from roots to apical meristems (Supplementary File S3). Relevant uses are for handcraft fabrication (which provides economic incomes), utensils directly used by households (domestic use), construction, and important ritual/ceremonial practices (Table 1).

**Table 1.** Palm parts and uses of *Brahea dulcis.*

| Part Used | Product | Use |
|---|---|---|
| Entire plant | Palmo | Ritual |
| | Palm | Ornamental |
| Inflorescence | Crowns for dances | Ritual |
| | Dried floral adornments | Ornamental |
| Fruits | Capulín/Soyacapolli * | Food |
| Apical meristem | Palmito | Food |
| | Infusion | Medicinal |
| Tomento | Cataplasm | Medicinal |
| Mature leaves (*Soyamatle* *) | Fuel | Domestic |
| | Fumigant | Domestic |
| | *Patchole* * | Domestic |
| | *Tlatepatchole* * | Construction |
| | *Tlatepatchos* * | Food |
| | Palm toy whistles | Domestic |
| | Green leaves | Ornamental |
| Leaf buds | *Acachiquehutle* * | Domestic |
| | *Tenate* | |
| | Aventador | Domestic |
| | Beehive | Domestic |
| | Strips | Handcraft |
| | *Cachache* ** | Domestic |
| | Capote | Domestic |
| | Crosses and Ramos | Ritual |
| | Hats | Handcraft |
| | *Mecapales* * | Domestic |
| | *Petate* | Domestic |
| | *Pixcalon* ** | Domestic |
| | *Soyacatle* * | Ritual |
| | *Soyate* | Domestic |
| | Strainer | Domestic |
| | *Tenates* | Domestic |
| | *Tecolpetes* * | Domestic |
| | *Tentematlat* */Temposha ** | Domestic |
| | *Tehiotzibtke* * | Domestic |
| | Thumbnails | Handcraft |
| | *Tlachpahuastle* * | Domestic |
| | *Tlaistechicone* * | Domestic |
| | *Xoxolochtli* * | Domestic |
| | *Zazca* ** | Domestic |
| | Jewelry | Handcraft |
| | Wrap | Domestic |
| Foliar bracts | *Coaxtli* * or pads | Domestic |
| Roots | Roots | Domestic |
| | Roots | Medicinal |
| Stem | Poles | Construction |

\* Word of Nahuatl origin; ** word of Nwiga origin.

The most important aspect of using this palm is related to cultural engagement and identity (Figure 3). Knowledge of palm weaving can start from an early age (6 years): "They teach me [his grandparents], in the beginning there are *panalitos* [referring to the weave of objects that resemble small honeycombs, which are the first thing that is taught when beginning to weave] [ . . . ] those are the first steps [ . . . ], that is why people from before, at 20 years old, already know how to make *petates*

and everything they want" (people from Zapotitlán Salinas Puebla). Among the *Xwja*, *B. dulcis* is considered a staple plant: "Palms are our life because with palm leaves, we make hats and we can get all we need to live" [53]. In different studies, interviews suggest that people remark not on the need of weaving, but their desire. As an elder *Ñuu savi* woman said, "My hands hurt, but I will not stop weaving because soon my little heart feels that it wants to weave" (San Pedro Jocotipac, Oaxaca).

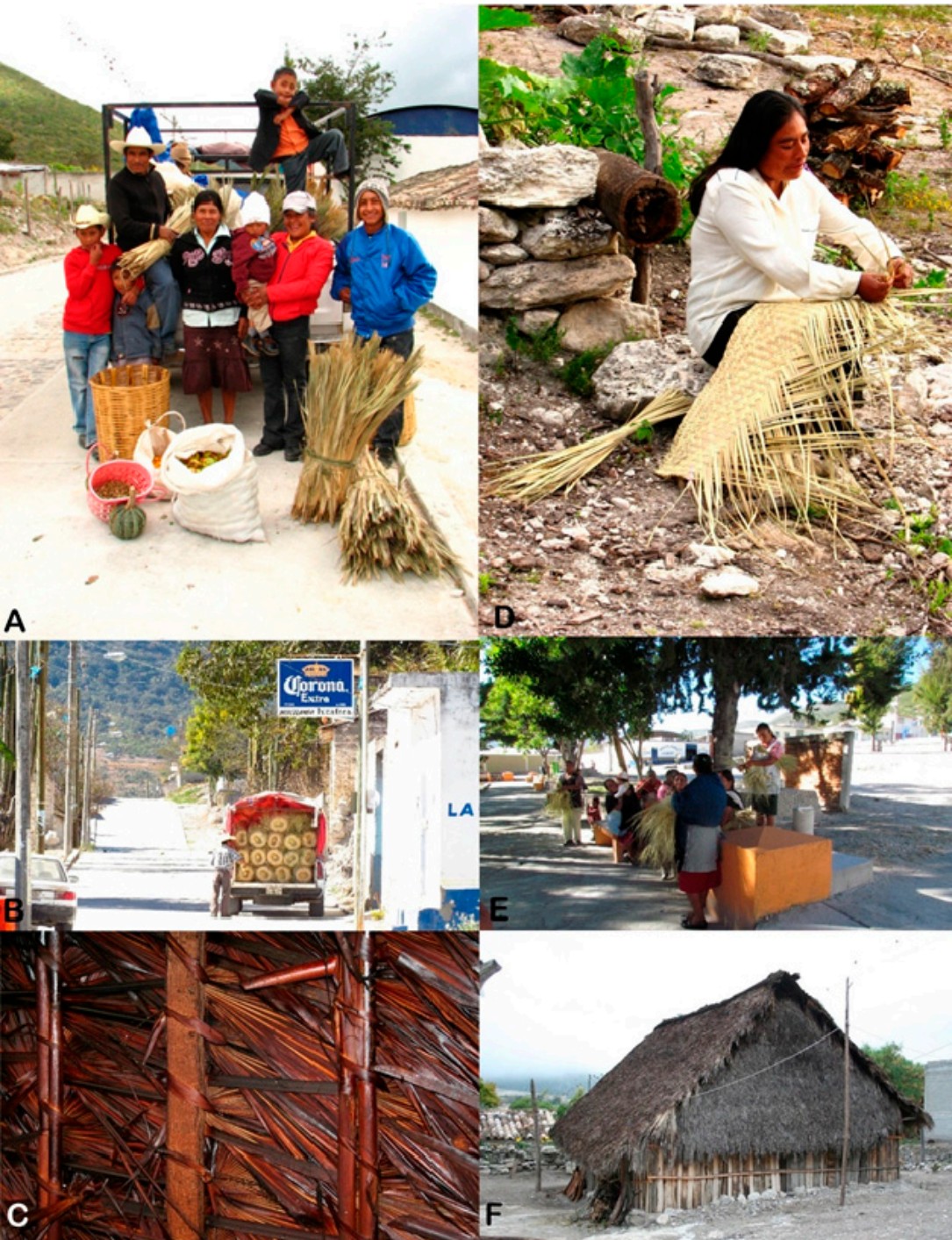

**Figure 3.** Cultural aspects of palm importance. (**A**) Interchange (*trueque*) of palm leaves for merchandise. (**B**) Hats ready to be transported to Tehuacán, Puebla. (**C**) Detail of a palm roof made with *Brahea dulcis* leaves. (**D**) Weaving of a *petate*. (**E**) Weaving as a form of social and intergenerational bonding. (**F**) Traditional house with palm roof at Santa María Ixcatlán. Photo credit for A, B, and E: Selene Rangel-Landa; C and F: Ignacio Torres-García; D: Erandi Rivera Lozoya.

Some renowned uses are those related with ritual and religious practices. When a person dies, the Nahua from Guerrero and the *Xwja* from Oaxaca weave *huaraches* (a kind of sandal) which the deceased person wears [20,28]. They believe that these sandals help the person to overcome the obstacles on their way to the afterlife [28].

The *Hñä hñü* people from Hidalgo weave a headpiece with palm in the form of a crown that bears a small cross; a story of the village tradition says that when a deceased man reached heaven, San Pedro asked him why he arrived in heaven with his head uncovered. The man then returned to earth to tell people that they must be buried with a headpiece [41]. Religious practices also include palm-woven pieces related to dances and *ramo*, a floral arrangement used during Easter which is blessed in a religious ceremony and provides protection. This is placed behind the front door of the house for protection by driving away bad spirits, and it is believed that when there is a storm or other strong atmospheric event, the burning of the *ramo* will bring calm (Zapotitlán Salinas, Puebla). In Santa María Ixcatlán, Oaxaca, the elders say that the ceremony of blessing the palm represents the sacrifices and the life of Christ. It is also a ritual praying for the palm and the woods, so that there is rain and they do not lose what nature gives. Palm leaves are also used with other plants in rituals known as *limpias* (healing ritual). This is because the palm is considered a purified plant and is used to heal people suffering from cultural diseases. It is also used in pre-Hispanic dances to summon rain or when asking for a good harvest season [28]; the entire palm is used as religious ornament in Zapotitlán Salinas, Puebla, and Huitziltepec, Guerrero during Holy Week (Semana Santa in Mexico) to decorate churches.

The weaving of objects for self-consumption, sale, or exchange also plays an important cultural role (Figure 3). For those weavers who have the possibility to move to commercial centers, the market is used to exchange products and obtain money [54]. *Brahea dulcis* is strongly traded in traditional markets across Central Mexico, such as Chilapa, Tulcingo and Tlapa, Guerrero) [55–57], Sahuayo, Michoacán [41], Ajalpan, Coxcatlán, Tehuacán, and Zinacantepec, Puebla and Cuicatlán, Huajuapan and Teotitlán del Camino, Oaxaca. Permanent transportation of palm leaves takes place via trucks, and therefore leaf pickers, handlers, handcrafters, and sellers are involved in commercial activities [20,58,59].

Hat production persists as an important economic activity [20,28,37], whereas other palm products are falling into disuse, likely as a result of precarious market conditions and decreasing demand. A decrease in the use of *B. dulcis* for thatching houses has been reported for the region of La Montaña de Guerrero [28,56]. In other places, roofing is already in disuse, such as in the region of La Huasteca [41]. In the region called La Meseta Purhépecha in the state of Michoacán, as well as in La Huasteca, the production of raincoats made of *B. dulcis* leaves, named *capote*, fell into disuse due to the introduction of plastic raincoats [41,60]. In the latter region, only older people are dedicated to palm weaving; young people have shown an increasing disinterest in learning palm weaving given that it is undervalued and poorly paid work [41]. According to people from Zapotitlán Salinas, Puebla, many domestic and agricultural uses of palm have been lost due to the introduction of plastic objects, which have replaced many utensils and tools previously obtained through palm weaving. The *zazca* (a large woven palm basket case for storing maize) and the *pishcalon* (a woven palm tool for cutting cobs) were utensils used in agriculture, as were the *cachache* for containing beans during their harvesting, and the *temposha*, a woven mask for preventing bulls from eating the green leaves of the milpa during agricultural labor (Table 1). As plastic objects started becoming available, they replaced palm utensils and progressively displaced them. Today, most agricultural needs are covered by plastic objects, and palm weaving persists as an activity which is mainly related to handcraft manufacture and commercialization.

The replacement of *B. dulcis* fiber by plastic strips for weaving is notorious in some regions. In the Tehuacán-Cuicatlán Biosphere Reserve, people recognize that *fibra* (as they named the polypropylene strip) is preferred by consumers because it lasts longer and can get wet without becoming spoiled. Furthermore, locals mentioned that what they like to weave is palm. In this biosphere reserve, the replacement of palm crafts by such plastic materials has caused the gradual decrease of the

traditional management of palms [36]. Nevertheless, it appears that palm weaving is far from disappearing. This activity remains relevant because of its economic importance. Income from the sale of woven objects is often the only monetary income, or woven items are bartered for food in local stores in numerous rural communities [20,28,37,61–63]. Weaving is done during spare time and while performing other activities, such as taking care of elders and children, and attending community assemblies. As it does not overlap with other productive activities, it is possible to maintain the activity despite its low economic return. Groups of people weave palm together in the early evening, when they have all concluded their day-labor. It is a kind of social activity, allowing conversation while working.

There is renewed interest in making small handcrafts such as earrings and thumbnails. These objects require less palm fiber but demand a greater investment of time for artisans, and their prices—in spite of being better than those received for the sale of traditional objects such as hats, *petates*, or *tenates*—are low considering the time invested, given that they require special labor time, since their elaboration requires more concentration than other traditional objects: in the artisans' words, they need to "sentarse a tejer" (sit down to weave), while traditional product weaving can be performed simultaneously with other activities, even while walking. Although the demands for earrings and thumbnails are low, there are organizational experiences in training and selling these products, such as the conformation of collective brands of palm products such as "Palmart's" [64] and "XULA Palma Artesanal" [65], which have reached national and even international markets by way of the Internet.

### 3.3. Economic Triggers

During the first half of the 20th century, the palm weavers of the region of La Mixteca received a daily income that did not exceed US$0.06 [66] ($0.16 Mexican pesos, based on an exchange rate of $3.60 Mexican pesos per US dollar). For the elaboration of a palm hat, the most highly demanded woven article, an adult weaver could require 12 h of work to weave 3–4 hats, for which they received US$0.08, from which they still had to subtract the cost of raw material. One hundred young palm leaves were priced at US$0.24 on average, which was enough to produce 12–18 hats. Therefore, *Ñuu savi* weavers had a net income of about US$0.02 for 12 hours of work [67]. Hat production was concentrated in regional markets and then distributed to companies that selected, bleached, and finished the hats for their later sale either in the country or abroad [52].

In recent years, one dozen ordinary palm hats or "en greña" (in the rough) cost US$0.8 (based on a modern exchange rate of $20.00 Mexican pesos per US dollar) [66]. From 2011 to 2015, the price paid for one hat was US$0.16 (US$1.92 per dozen), and they were then moved to industrial plants where they are cut, dried, bleached, sewn, and ironed, increasing their price value by up to 1000% [58]. Although this low return has not changed for decades, the weaving and selling of palm objects continues since it is the most important or the only source of monetary income for numerous rural households—income that pays off food and other products not produced by the families [20,28,57,61,68]. In 2001, it was estimated that palm was an essential economic source complement for more than 50,000 peasant families of Mexico [51]. During this year, in the locality of Santa María Ixcatlán, Oaxaca, the price for one hat—which takes around three hours of work—was US$0.33 (based on an exchange rate of $12.00 Mexican pesos per US dollar). Despite monetary compensation for this activity being inequitable, the sale of these objects has provided crucial economic income for sustaining families since early colonial times [69].

The trading of palm leaves between localities is important for sustaining palm weaving at the regional level. *B. dulcis* is not equally available in all palm weaving-dependent rural communities. These communities—for instance, San Luis Atolotitlán in Puebla—are supplied by regional hoarders who periodically visit the rural localities [62]. One way in which these hoarders obtain the leaves is by exchanging products such as wheat, maize, or fruits, for palm leaves with harvesters in towns where *B. dulcis* is abundant, as is the case in Santa María Ixcatlán, Oaxaca [20].

### 3.4. Brahea dulcis Traditional Management

Traditional management refers to several non-modern man-made practices conducted at several scales of social organization to enhance the availability of natural resources. The following practices are regarded as traditional—each is related to one category of management (gathering, toleration, protection, promotion, cultivation) [2], and holds a particular purpose—these practices also entail ecological implications that have spatial outcomes (Table 2 and Figure 4).

### 3.4.1. Gathering

Gathering involves the harvesting of fruits, stems, foliar bracts, and leaves. Here, we refer to the harvesting of young and mature leaves, given that this is the most economically relevant palm product and the best-documented activity. In the case of young leaves or *cogollos*, the annual frequency of harvesting events can range from one to eight, or even every week. The gathering can be conducted by people that both harvest and weave the palm leaves, but there are people that only harvest leaves to sell them [44,61]. This is usually carried out by family leaders, frequently men, as the main activity of a journey, or practiced opportunistically during other activities, such as firewood gathering or pastoralism practices [20]. Although harvesting is possible throughout the whole year [61,62], people in some communities prefer to harvest palm during the wet season because young palm leaves, which are softer and easier to handle, are more abundant than compared to during the dry season [57]. Harvesting and weaving are also related with patterns of production, occupation, and necessities. During the dry season, agricultural activities are less demanding, and people can spend more time gathering palm and weaving [20]; people can also dedicate more time to these activities when there is a special need for monetary income, as is the case when there are bad years for agriculture and production is low.

Decisions and even regulations about when to carry out the gathering of palm leaves are based on moon cycles, which are related to leaf quality and duration, as similarly reported for other palm species [9,12]. There is a consensus in preferring the crescent moon until a few days after the full moon for harvesting; it is believed that, in this way, the palm and production of leaves is not negatively affected, and otherwise leaves are prone to being eaten by insects, becoming fragile and rotting because of excessive water [5]. In some communities, if this criterion is not respected, people can be penalized [62].

Palm leaf extraction is mainly practiced in agricultural areas where palms have been left standing and in palm stands—most probably a secondary vegetation derived from management [37,62]. Harvesting can occur at distant sites from human settlements where *B. dulcis* is found in natural vegetation, secondary forests, and pasture lands [41,44,62,70]. Harvesting on low-stature individual palms is oriented toward larger and immature leaves, while in arboreal palms, this is oriented to mature leaves, foliar bracts and fruits. There is little information related to the time invested in transfers to harvesting sites. In the documented records, transfer times can range from 15 min to 14 h [44], which translates to a ~400 m to 21 km round trip. The harvesting of young and mature leaves is conducted without felling the palm, and it is socially agreed that felling should not be practiced unless planning to use the entire palm or stem; it is frowned upon to do otherwise [70]. The harvesting of small individuals is conducted with a machete (bowie knife), and taller ones are harvested with a purpose-made cutting tool, which is a stick (that can be from *Arundo donax* L.) with a blade on its tip; people also use this tool to make incisions on the stem like steps of a staircase, making it easier to climb the palm and reach the leaves. Harvesting consists of cutting mature or young leaves that have not yet unfolded and cutting is done with care to avoid harming the new leaf buds [20]. Valderrama et al. [44] recorded average harvest intensities of 471 leaves per harvest event, with a range of 200–600 leaves, and Casas et al. [58] estimated gathering ranges between 300–400 *cogollos* per day of labor.

**Table 2.** *Brahea dulcis* management implications.

| | | | Purposes | Ecological Implications | Spatial Implications | Spatial Outcome |
|---|---|---|---|---|---|---|
| **Management practices** | **Harvest** | *Intensive* | Promotion of small stature individuals and increase vegetative propagation | Promotion of colonial life habits with same genetic information (ramets), small stature and short leaves | Promotion of palm stands with accelerated vegetative propagation of colonial life habit, which can cover areas of different sizes | *Manchonera* |
| | | *Restrained* | Maintain high stature individuals and sexual propagation | Maintain individual life habits and promotes genetic diversity | Promotion of palm stands of high stature individuals, which can cover areas of different sizes | *Soyacahuitera* |
| | **Tolerance** | *Selective felling* | Leave useful palm individuals standing | Palm individuals can benefit from elimination of competition | When lands left under rest or abandoned, palm individuals can easily propagate and cover cleared areas | **Propagation** of palms to abandoned areas |
| | **Promotion** | *Burning* | If incidental, reduces plant competition by non-fire resistant species | Palm propagation by regrowth of resistant individuals | Burned areas prone to be covered by palm stands resulting of palm fire resistance and elimination of competition | **Propagation** of palms to burned areas |
| | | *Cleaning* | Stimulate growth and increase productivity | Improves individual's fitness | - | - |
| | | *Weeding* | Eliminate competition and promote nutrients reassignment | Improves individual's fitness | May prevent the growth of palm stands by removing seedlings | **Deceleration** of palm stand growth |
| | | *Grounding* | Improve substrate conditions | Improves individual's conditions | - | - |
| | **Protection** | *Harvest restriction* | Control harvesting | Prevention of over-harvesting | May prevent the expansion of palm stands by controlling harvest intensity | **Preservation** of certain areas or places |
| | | *Prohibitions* | Regulate periods and places of extraction | Prevention of over-harvesting and protection of particular places | Some places may be under protection regulations in order to conserve them | |
| | | *Nature observation* | Respect natural cycles | Collective benefit by understanding of management impacts | - | - |
| | **Cultivation** | *Transplanting or sowing* | Ornamental purposes | Palms presence beyond its natural dispersion limits | It could conform large areas under cultivation if were intensively sowing or transplanted | **Conversion** of natural places to crops * |

\* Not seen. Bold: emphasize.

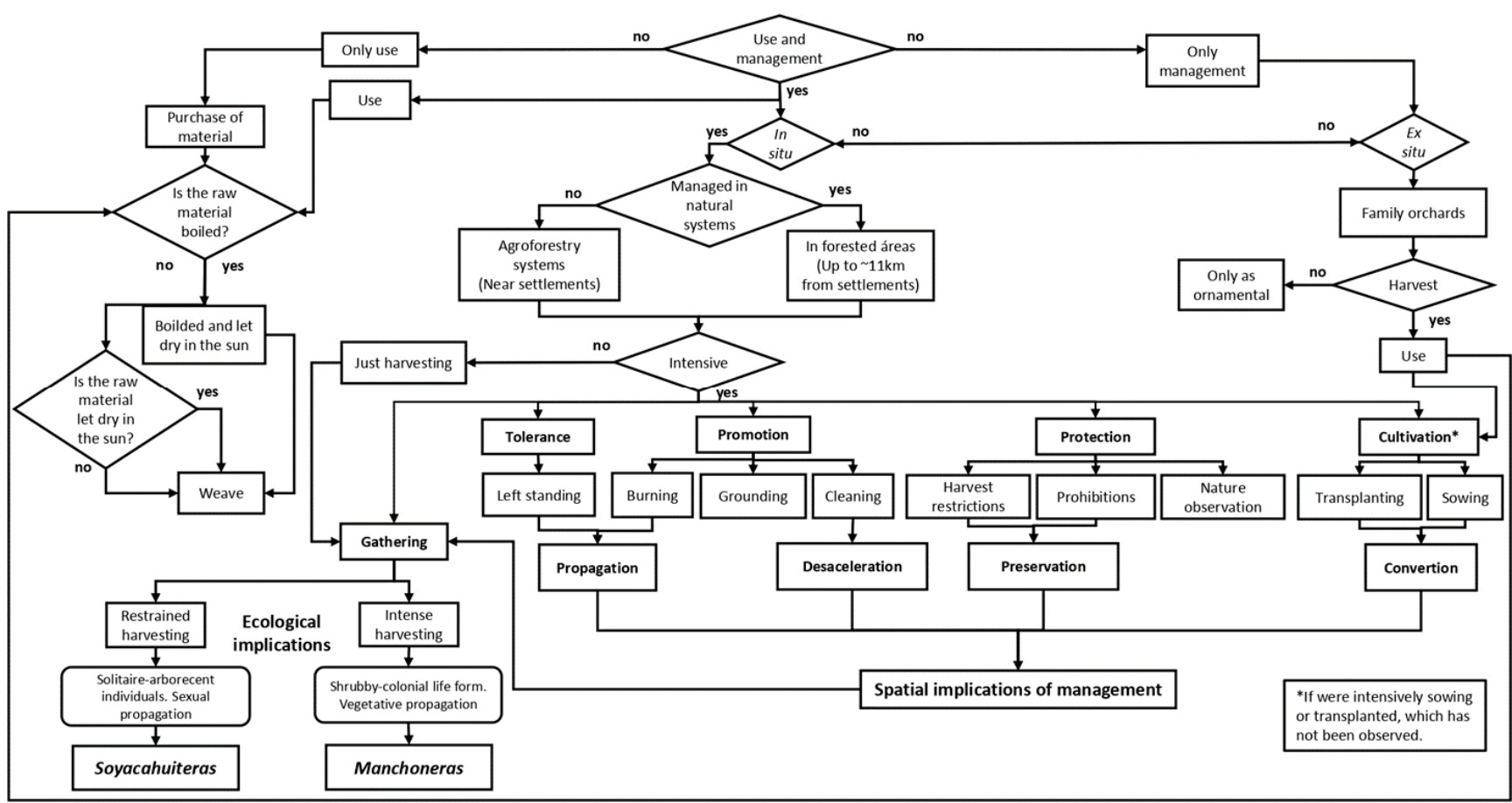

**Figure 4.** *Brahea dulcis* management flowchart.

### 3.4.2. Constant Harvesting

This practice is related to the maintenance of low-stature individuals, growth retardation of the stem, and an increase in vegetative propagation. Continuous pruning stimulates the increasing availability of stem and leaf buds for harvest from *manchoneras* [37]. A *Hñä hñü* harvester from Guerrero mentioned that "it is necessary to harvest constantly the leaves in order to avoid stiffness so that leaves can be used for selling" [44]. Nevertheless, peasants associate constant harvesting with a reduction in leaf size [37,55], which is an unwanted consequence given that the availability of larger leaves is reduced. In the case of *soyacahuiteras*, the harvesting of leaf buds is never practiced in some localities for these reasons (Restrained harvesting on Figure 4) [37]; this is a topic that demands further research due to the lack of conclusive information [55,63]. On these palm stands, harvesting is oriented toward extracting primarily dry leaves and foliar bracts, and practices that could risk the permanence of these resources are avoided [28,37].

### 3.4.3. Selective Felling

This activity refers to the act of letting certain valued plants stand when clearing a vegetation area and removing plants that are not useful. When land is cleared for farming in agroforestry systems in numerous regions of Mexico, the usual practice is that useful trees are kept standing [71–74]. This practice has been reported for palm species in northern and southern Mexico, as is the case for several species of *Sabal* and *B. dulcis* [20,22,29]. On plots cleared for cultivation where *B. dulcis* individuals are maintained, these benefit from management practices dedicated to the care of the *milpa* (traditional Mesoamerican polyculture of maize, beans, squashes and other species) [28,37]. The main reason for leaving the palm standing is because of its use and the strength of the roots affixed to the soil [37]. Letting palms stand can allow the initiation of a new palm stand; the elimination of shading and competition with other plants allows the palm to spread on the farmland during fallow periods [28].

### 3.4.4. Burning Practices

The practice of burning an area is related to agricultural activities. Fire events may not be incidentally used for the benefit of palms but may involuntarily promote palm dominance given the fire resistance of *B. dulcis* [28]. It also has been proposed that palm stands of this species in La Mixteca are a product of deliberate fire and extensive livestock [1,58,75,76]. Involuntary promotion of this palm can derive from burning as a common practice in slash-and-burn agriculture. These induced fires damage most herbs, shrubs, and trees and are the reason why a Nahua user from Guerrero said "when the mountain burns, the other season is full of palm" [28], as palm leaves burnt by fires fall out and sprout anew. Fire use is avoided in the *soyacahuiteras* from La Montaña de Guerrero region, since these can easily be affected or destroyed [20,55]; also, fire was recorded to accidentally benefit *manchoneras* palm groves in this region [37,55]. Uncommonly, some people rely upon burning areas within their plots to eliminate dry leaves and herbs around palms.

### 3.4.5. Practices of Special Care

There are some management practices targeted towards improving palm development. Cleaning consists of removing dry leaves from the stem. According to Illsley et al. [37], this will prevent leaves from becoming a shadow for the scarce rain on these regions, allowing humidity to reach the root area. The dry material is then minced and left at the base of the stems, helping the plant to conserve moisture and returning nutrients to the soil, which favors stem growth. Tiller thinning or *deshije* consists of removing vegetative shoots in order to reduce competition for resources. This also promotes nutrient reassignment and results in increased stem growth and foliar production. This practice also includes weeding for maintaining monospecific stands [55]. Another important management practice consists of improving soil conditions for palms by bringing soil from other places, which is named

"arrime de tierra", or grounding. This practice, as well as sowing, is only carried out in orchards, home gardens, or on newly open farming lands [37].

### 3.4.6. Social Organization and Regulations related to Palm Use

In indigenous localities, where land is conceived and legally sustained as social property, it is common that all community members have equal rights to access resources in the communitarian territory, but decisions related to land use rights are discussed and defined in communitarian assemblies; thus, management is carried out based upon customs and traditions (uses and practices are regulated by communities as traditional forms of appropriating resources). In Santa María Ixcatlán, Oaxaca, all members of the community have the right to harvest leaves for their own use; commercialization of palm leaves is allowed between community members. In this way, palm can be purchased by people who cannot harvest palm themselves, such as the elderly and sick people who need to weave the palm to survive. People who arrive from other places offering products or merchandise—such as fruit, maize, wheat, clothing, and home utensils—can receive payment through barter with palm leaves [20]. In some communities, the harvesting of palm leaves to sell outside the community is prohibited as a precautionary regulation [61]. In localities where there is a regime of land tenure for small landowners, the owner of the plot and their family are the only ones allowed to access the resources, but can establish selling or interchange agreements, or even freely allow people to make use of resources inside their property [37].

Rules may vary among communities but are oriented to avoid contested situations. If there is a decrease in arborescent palms from which dry leaves are harvested, rules and actions are agreed upon to favor the presence of these palms, such as not allowing the cutting of stem and leaf buds or extracting foliar bracts. There are also implicit rules that are transmitted intergenerationally and are respected because not doing so is socially frowned upon. These include not cutting the palm stem, careful harvesting to avoid damaging the apical meristem and compromising palm welfare, and not harvesting more leaves than necessary [20]. Issues related to respecting moon cycles for harvesting are sometimes implicitly followed, but if these are not respected in some localities, may result in economic sanctions [62]. In Hidalgo, some *Hñä hñü* communities believe that palms do not need special care, given that they grow on their own, and it is only necessary to harvest them so they do not become "stiff" and can keep producing good-quality leaves.

### 3.5. Ecological and Ecogeographical Implications of Management

The constant palm management has impacted ecological aspects such as species phenotypes, abundance, distribution and sympatric biodiversity. As these processes are derived from dynamic interactions with other species and may be related to the plant response to environmental factors, which affects their abundance and distribution, they can be defined as ecological [77]; however, since management practices are performed in a geographical space, there is also a spatial dimension of the ecological implications (Figure 4), which have their clearest imprint at the landscape level.

### 3.5.1. Ecological Implications at Population and Community Levels

According to Quero [38], the *Brahea* genus population has increased in abundance despite its intensive management. The phenotypic plasticity of this species allows it to respond to intense harvesting by developing a colonial life habit that can hold up to 22 stems covering areas of 24 m$^2$ [78]. This clonal spreading appears to facilitate resource uptake in modified ecosystems and the rapid clonal propagation of seedlings [79]. The hybridization of *B. dulcis* with other species of this genus, such as *B. calcarea*, is another factor that increases morphological variability [80]. Extensive palm stands of *soyate* have been documented for several regions of Central Mexico [37,58,59].

Leaf production varies widely and is apparently little affected by harvesting [41,55,56]. The annual leaf production per individual is in the range of 5–20 leaves [36,37,61]. Pulido et al. [41] compared leaf production amongst the different arborescent palms of the Americas and showed that *B. dulcis*

had the highest leaf-production rate. According to previous studies of the authors, *B. dulcis* produces numerous leaves within a relatively short lifespan, making it ideal for management and harvesting.

During one-year studies [36,41], the harvesting of *B. dulcis* did not show significant effects on individuals and the population death rate. In most studied cases, leaf harvest has not been shown to have a significant impact on palm leaf production or the demographic parameters of *B. dulcis*. This is a similar pattern as that found for other palm species used in Mexico, such as *Sabal mexicana* and *S. yapa* [40]; although for *B. dulcis,* peasants had observed a decrease in the height of individuals which is associated with leaf size reduction [37,56,57]. The leaf harvest and management practices of *B. dulcis* have an important influence on plant morphology and in maintaining a reservoir of genetic diversity. Management has derived stands of secondary origin with a high abundance of palm known as *palmares*. In several locations, people reported the recognition of two types of palm groves; given the sharp difference between the two phenotypes, it was first believed that they corresponded to two variants of *B. dulcis* [73], when in fact they are two morphs of the species [81], apparently derived from the manipulation of phenotypic plasticity by centuries of management [28,37]. This is supported by recent studies that found a significant and negative effect on adult palm height derived from leaf harvesting [79], and have been registered in several places of central-southern Mexico [37,41,44,55,56,61,70].

These two palm groves are distinguished by differences in height, propagation, and developed life habits (Figure 2). Low-growing palms stands, of about 1.5 m tall, reproduce predominantly by vegetative propagation and are characterized by caespitose growth-forming colonies known as *manchoneras* [37]. Leaf buds are intensively harvested from this kind of palm stand and elaborated into hats and diverse utensils (Table 1). The constant cutting of leaf buds maintains the low size of these palms and promotes the growth of several stems of small size, which facilitates harvesting and confers a shrubby appearance (*manchoneras* on Figure 4).

Stands dominated by palms of more than 6 m in height are known as *soyacahuiteras*. These are characterized by vigorous sexual reproduction and solitary growth habits. This kind of palm grove is maintained mainly for harvesting foliar bracts, fruits, and mature leaves. These leaves are used for thatching traditional roofs of houses. Foliar bracts are used for the manufacture of a special cushion for loading donkeys in different parts of La Montaña and the Tehuacán-Cuicatlán Biosphere Reserve. Fruits are eaten as candy [20].

Palm groves made by these two *B. dulcis* morphs are subjected to different management practices according to what is needed to maintain their characteristics and, through management, a *manchonera* can grow into a *soyacahuitera* [37]—a process recently documented in San Francisco Cotahuixtla, Oaxaca [47].

Other effects of palm management on ecosystems have been little studied or described, such as the effects of the management of palm groves on biodiversity. Although the promotion of *B. dulcis* on natural vegetation derives from biodiversity loss, there is a substantial proportion of plant diversity maintained within palm groves which is not retained in conventional crops. In Santa María Ixcatlán, the surrounding natural vegetation in which the palm is naturally distributed has recorded diversity values of H = 1.516 for the *Mexical* [3], a kind of *Quercus* scrubland; H = 1.47 for a *Quercus* forest; and H = 1.28 in a *Juniperus flaccida* forest, while palm groves retained a diversity of H = 0.827 [20].

### 3.5.2. Ecogeographical Implications

The management of palms occurs in forest, agroforestry or agricultural areas. It is common to find palm individuals used as living fences, terrace borders, or deliberately left standing in agroforestry systems of Mexico, given that they are appreciated for their fruits and leaves [71]. Palm management is also carried out in forest areas, where management practices have the potential to influence and transform plant communities, resulting in important modifications to plant distribution, population, and diversity parameters at the landscape level [81]. These plant formations in which *B. dulcis* is the dominant floristic element are believed to be deliberate human-made stands derived from management

or disturbance, particularly from induced fires and the clearing of vegetation for agriculture. This is supported by recent studies that found that management has a significant and positive effect on palm density on managed stands where fire routines are performed, which is associated with slash-and-burn agriculture [79]. The *B. dulcis* management flowchart (Figure 4) indicates those management practices associated to spatial outcomes. The action of palms left standing on cleared or abandoned lands can set the mark for palm individuals to facilitate their propagation [28].

Erosion also plays a part in palm grove establishment. *Brahea dulcis* is highly tolerant to soil erosion and degraded environmental conditions, so it can be easily established and can endure on abandoned farmlands and deteriorated terrain [61,62]. The people from the La Montaña de Guerrero region who manage the palm commented that *soyate* is resilient to the cutting off of leaves, sprouts and stalk. Even then, the plants produce shoots for up to ten years. It sprouts very well after fire: "it endures the trampling and browse of animals, and endure[s] the drought" [38]. As a plant that is left standing when clearing for agriculture, when the land is fallow, *B. dulcis* freely propagates on these open lands [5]. On forested areas some people promote the propagation of the species by deliberate fires such as in the case of La Montaña [55,56] and La Mixteca regions [58]. As a fire-resistant species, slash-and-burn practices and the deliberate burning of natural vegetation can promote *soyate* palm distribution and density by increasing areas for colonization, reducing competition, and fostering an increase of recruitment through sexual and asexual mechanisms [79]. Quero [38] states that *Brahea dulcis* stands are favored by anthropic disturbance and increased populations of this species. On the other hand, there are certain practices that avoid palm stand expansion. The cleaning of tillers and harvesting restriction and prohibition acts as decelerators of palm stand growth by regulating the presence and development of offshoots, as well as by regulating places, periods and intensities of extraction. All the management practices have spatial implications that differ in their outcomes (Figure 4).

Palm stands of *Brahea dulcis* are commonly found in the upper Papaloapan Basin, Central Chiapas, and the state of Oaxaca State (at southern Mexico), Barranca de Metztitlán, Hidalgo, the Tehuacán-Cuicatlán region, the Balsas river basin (central Mexico) and from the mountain range Sierra Madre Oriental to the southern Tamaulipas State (northeastern Mexico) [3,41,75,76].

## 4. Discussion and Concluding Remarks

### 4.1. Traditional Management: Insights for Sustainability

*Brahea dulcis* ought to be regarded as the palm species with most widespread cultural relevance in arid to sub-humid regions in Mexico. The cultural and economic importance of *soyate* palm, as other traditionally managed resources, has been developed through generations of interactions [82]. Since management is place-based, it has triggered an intermingling of ecological and cultural landscape configurations. Currently, management practices reported for *Brahea dulcis* in the country include all forms reported by Gonzalez-Insuasti et al. [2], such as gathering (selective harvesting), toleration (leaving standing), propagation (the promotion of vegetative reproduction through pruning and burning), protection (cleaning, grounding, weeding, social conventions), and cultivation (ex situ planting on family gardens). This is a gradient of management intensity that does not resemble steps of a sequence and does not exclude them from each other [1]. They conform to a broad spectrum of management that is implemented at different intensities, which varies regarding the degree of sophistication, number of people involved, and number of management practices performed, which relates to its ecological and cultural importance [2]. Based upon the ecological adaptations, it is here stated that *soyate* palm might be harvested sustainably as a profitable resource [53]. Nevertheless, over-harvesting must be carefully avoided since it could result in a decreased leaf size, which has implications for its use, given that certain leaf sizes are needed for specific resource uses [37,55]. Studies must be oriented to define thresholds of harvest, in order to define harvesting dynamics that can ensure the maintenance of an optimal leaf size [45]. *Soyate*'s traditional management promotes sufficiently frequent seedling recruitment of different genotypes, thus maintaining high genetic

diversity; the renewal of palm individuals by slash-and-burn events may even enhance genetic diversity, which indicates that current traditional management practices on these sites may not only preserve the resource, but also its capacity to cope with environmental changes [79].

The high number and effective management practices relating to *Brahea dulcis* use (Table 2) are particularly outstanding when compared to other palms of South America. According to a recent review by Bernal et al. [83], at least 22% of used palm species have no record of management, 55% are subject to one or two, and only 7% of palm species are subject to five to eight management practices. *Brahea dulcis* management involves sophisticated practices that integrate social organization and nature observation; unsustainable practices have been avoided by respecting natural cycles and precluding unnecessary damage, such as the felling of the tree palms, which has been reported in South America for the harvesting of palm leaves and fruits of *Astrocaryum chambira*, *A. standleyanum*, *Aphandra natalia*, *Mauritia flexuosa,* and *Oenocarpus bataua* [83]. Furthermore, practices benefit *B. dulcis* by promoting resource availability and competition elimination, which results in increased palm density [28,37] and also the permanence of palm stands [36,72]; however, information relating to the impact of palm stand promotion on loss of biodiversity must still be assessed, since the dynamics of accumulation of organic matter, nutrient cycling, and soil maintenance are drastically altered in palm stands with respect to primary vegetation. Effects on soil and hydric dynamics in palm groves that are still poorly known need to be assessed. Despite the relevance of *Soyate*, a further assessment of other palm species in Mexico is needed, such as for the *Sabal* species which represents an important resource in the warm and humid regions of southeastern Mexico [29,39,40].

From the ten recorded management practices, at least six imprint landscape configurations (Figure 4). Selective felling and burning is related with the propagation of palms stands, resulting in the opening of new palm areas or the expansion of established palm stands; weeding is related to the deceleration of palm stand expansion due to the control of vegetative growth; preservation of certain areas due to protection is carried out by social organization of users who have constructed harvesting restrictions or prohibitions in particular places; and although these are not yet seen, the massive transplanting of palms or sowing in certain areas could lead to the total conversion of natural places to croplands. Thus, use and management may have a profound influence on palm attributes and distribution.

Selective felling, slash-and-burning, weeding, protection of certain areas, and the massive transplanting of palms are distinguished as the most relevant ways to transform the region into a cultural–natural mosaic landscape. We state that the recognition of spatial implications of these practices is necessary to assess the impacts and benefits of management, given that socio-economic and environmental pressures can trigger palm stand expansion [1,29,84]. Long-term trends are yet to be ascertained, and so far, they have been difficult to unravel since socio-economic drivers are changing constantly [38]. The authors strongly argue that land use planning must be oriented towards regulating and preventing the expansion of *soyate* palm stands. This is especially important to reduce the impact upon other native—and highly biodiverse—vegetation types, which are often diminished and disturbed to favor *palmares*. If the creation of new palms stands is necessary, this could be promoted on degraded sites and under situations of degraded soil, where the palm could play an important role in preventing increased soil erosion [55,56]. According to managers of this species, the *soyate* palm plays an important role as a pioneer plant in successional processes, and its (fascicular) roots favor its ability for trapping soil and prevents its erosion [41,55]; it also favors the formation of organic matter and help in microclimate improvement for the establishment of other plant species [79].

Although the concept of space is implicit in traditional management, the way this interacts has been largely overseen despite the broad-scale implications of historical human behavior patterns relating to in situ resource management [85,86]. In Mesoamerican ethnical groups, as in most other ancient cultural regions, people integrally manage their territories. In our study, a special focus on the landscape level proved to be useful to understand the man–nature relationship as a baseline for setting sustainable management [85,87,88]. Hence, precise traditional management practices, performed at

local scales, (dis)favor the occurrence of certain phenotypes or species with important implications at the landscape level [89]. Wide-ranging practices, in order to increase the relative abundance of certain species at the expense of others, may result in large-scale patterns of transformation, which have been termed cultural niche construction [90] and landscape domestication [85,91]. These transformations of natural spaces are inherited among human groups in the form of ecological-and hence geographic-inheritance [92], and the permanence of transformed spaces is accomplished by the continuity of management practices through time.

Understanding the dynamics of the traditional management of non-timber forest products is complex because several factors are involved. In Figure 4, we propose a general view of the management of *B. dulcis*. The flowchart shows the set of decisions that are made relating to this resource, displaying actions that are translated to specific outcomes. The degree of participation and intensity of management depends on the involvement or not of managing actions. Some people only use the resources; thus, they do not actively participate in actions that benefit or drawback the resource or the natural context of it, so they may be unaware of harvesting actions that are detrimental. On the other hand, people that are actively involved in management make deliberate decisions to derive certain resources (e.g., the managing of a *soyacahuitera*) or to reach certain purposes (e.g., promotion), and some of these actions have explicit spatial outcomes (e.g., propagation of palm stands) while others do not. These management actions also occur in different contexts (agroforestry systems and forest areas), and consequences of management will differ among them. For the case of *soyate* palm, attention should be paid to practices that involve spatial outcomes on forest areas, given that natural vegetation may be eliminated as a consequence of palm stand propagation. A balance of the benefits and harms of this should be estimated before any actions are taken.

### 4.2. (Un)weaving a Sustainable Future

Despite the multidimensional relevance of *B. dulcis*, the trend of disuse of activities and products has been driven by the introduction of plastic and tools that replaced palm-woven objects [28,33,41]. In addition, this trend of disuse grows due to the low remuneration that weavers can expect from their work. As a result, young indigenous people become disengaged from learning this activity and focus on searching for better life opportunities outside their communities [36]. This, in many cases, implies permanent migration and family fragmentation, and the abandonment of agricultural activities with important repercussions for basic and sensitive subjects, such as the endangerment of self-supported food systems and community disarticulation [93,94]. Another discouraging issue concerns palm harvesting regulation established in the NOM-007-RECNAT-1997 (Mexican normativity that regulates the extraction of plant parts). Regional authorities regulate palm leaf harvesting through extraction licenses that are expensive and difficult to obtain, which prevents the accessibility of many people to legal extraction. Some authors state that regulations are enforced in an excessive way given that demographic studies have demonstrated that palm leaf production can endure harvesting [36,63], and through management, the genotypic diversity is maintained [79]. The revision of the current normativity on palm use is necessary in order that the imposed regulations are founded upon scientific studies and oriented toward protecting resources and the involved human groups.

### 4.3. Further Research

Ecological features are insufficient to depict the sustainable management implications of the *soyate* palm. Socio-economic, cultural, ecological and geographical factors need to be regarded holistically to eventually develop a sound sustainable management scenario for the *B. dulcis* socioecological system. On the socio-economic side, this represents an essential part for thousands of rural families—in some cases, as the only way to gain monetary income and as a currency in communities where income is limited [20,28,68]. On the cultural side, it participates as an important part of ceremonial and religious events of spiritual significance; moreover, it is considered as a substance that sustains day by day: it strengthens the cultural identity of people by the continual practice of shared activities and knowledge,

which bonds people with their ancestors and roots people to their territory [95]. It serves as a means of family integration, promoting intergenerational links through the process of teaching and weaving, and of intercultural recognition and social articulation through commercialization as a process that unites and reinforces "community identity" [95,96]. Ecological and geographical aspects have been thoroughly explained before. We regard our contribution as an effort to set a baseline for further efforts which may eventually drive traditional *soyate* management into a sustainable program.

Five research lines have been considered as critical for favoring the sustainable management of *B dulcis* and its habitat.

Participatory communication should be encouraged to engage stakeholders into constructive packs and agreements to set and implement regulations and rules for sustainable management. Attention may be focused on the spatial implications of management and disturbing actions as well as on soil erosion, soil fertility, sympatric biodiversity and hydrological dynamics. Also, the geographical determination of current palm groves and the evaluation of factors linked to their presence can support estimations of future conditions, as well as the extent and distribution of palm stands.

A holistic socio-economic, cultural, ecological and geographical scientific approach for *B. dulcis* management and its habitat should be encouraged to understand its role as an umbrella species for the sustainable management of non-timber forest products. Actions must depart from comprising economic benefits to peasant communities while promoting ecosystem conservation.

A fair appreciation of palm weaving should be fostered, in which products must be regarded as social and environmentally sound products. Palm products, contrary to plastic ones, have a low carbon footprint, enhance cultural identity and reinforce the long-term conservation of environmental services. It is important to acknowledge that palm-weaving techniques represent knowledge built through generations, beginning in antiquity; technique innovation and product diversification have permitted their continual use until the present. For that reason, palm objects are a repository of knowledge and of cultural value.

Regional-scale studies in order to understand the heterogeneity of management across ethnical groups and landscapes should be conducted. Proximal and underlying triggers and drivers of management must be identified in order to favor sound regional sustainable management, so that one human settlement does not jeopardize the neighbor's activity. This is especially relevant to gain insight into the different management strategies and to avoid contested situations within the cultural mosaic comprised in the region, including Central America.

Finally, researchers should review and document the historical importance and ecogeographic inheritance of *Brahea dulcis* management based upon codices and archives. Oral and empirical sources of information suggest the long-standing tradition of *soyate* management in precolonial times [97]. This information is needed to reveal the historical and cultural importance of non-timber products in large parts of Mesoamerica and other regions worldwide.

**Supplementary Materials:** The following are available online at http://www.mdpi.com/2071-1050/12/1/412/s1. Supplementary File S1: Bibliographic query results. Supplementary File S2: Mexican palm species and their uses; updated list of *Arecaceae* species native to Mexico categories and parts used of Mexican palms. Supplementary File S3: Palm parts used and weave objects.

**Author Contributions:** C.X.P.-V. was the leading author who conducted all of the research as part of her PhD studies; she further contributed to the systematization, data analysis, and writing of this paper. A.I.M.-C. suggested the elaboration and structure of the present manuscript; jointly with A.V. they are PhD advisors, and constantly guided the research and reviewed the manuscript. S.R.-L., J.B., J.C. and A.C. have been working with Brahea dulcis in ethnobiological, ecological, cultural and conservations initiatives for over 20 years; they also revised, commented on, and improved the manuscript. All authors have read and agreed to the published version of the manuscript.

**Funding:** The first author acknowledges CONACYT for their support with a PhD scholarship, the PAEP program for the economic support for fieldwork received through the Centro de Investigaciones en Geografía Ambiental, UNAM. To the DGAPA-UNAM for supporting the PAPIIT IN200417 project and CONACyT for financial support of the project A1-S-14306. This study is part of ongoing research of the thematic network of the Non-timber Forest Product Network (Spanish acronym, RPFNM) of CONACyT.

**Acknowledgments:** The author thanks Biol. Fernando Reyes, director of the Tehuacán-Cuicatlán Biosphere Reserve for all the support granted for the ongoing research in the reserve relating to *Brahea dulcis* management. Thanks go to all the people who kindly shared their knowledge, and especially Maurino Reyes from Zapotitlán Salinas Puebla for his tireless work as a promoter of the natural and cultural heritage of his region, and to Ignacio García-Torres and L.C.A. Erandi Rivera-Lozoya for sharing their photographic work for this document. We finally thank Base de Datos Etnobotánicos de Plantas Mexicanas, Jardín Botánico, Instituto de Biología, UNAM (BADEPLAM) and the biologist Laura Cortés Zárraga, for sharing information regarding palm use in Mexico. Final English editing was conducted under the guidance of the service provided by the present journal support system and the Academic Writing Office (Susana Kolb) of UNAM.

**Conflicts of Interest:** The authors declare no conflict of interest.

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
