# Peer review of "Ecological, Cultural, and Geographical Implications of Brahea dulcis (Kunth) Mart. Insights for Sustainable Management in Mexico"

_sustainability, doi:10.3390/su12010412_

Round 1

Reviewer 1 Report

The quality of the article became much better after resubmiting. The authors took into account the main remarks of reviewers. I reccomend to accept this article in present form.

Author Response

Thank you

Reviewer 2 Report

The authors have improved the manuscript, particularly by reducing its length and placing some content in supplementary materials. 

The English grammar has been improved, making the article more readable but still requires significant editing. For example the opening line in the abstract:

"Palm plant species are recognized as significant livelihood for rural communities worldwide." Palms are not a livelihood - but rather provide a resource that support livelihoods.

Author Response

It has been reviewed by the English correction service of the journal.

Reviewer 3 Report

This study addresses a study regarding the current status and the sustainable management of a palm species (Brahea dulcis) in Mexico. The aims and objectives of the paper fit in the editorial policy of this journal. I think the authors have made a worthy effort, and the manuscript is very promising. However, despite its evident strengths, my overall recommendation is that the manuscript should be not accepted now in its present form. My comments are listed as follows:

The authors should clarify the term “survey” in the abstract The term “sustainable management” appears in the title, and, profusely, in the last Section (“Further research”). For the reader, it is not entirely clear if the traditional and/or the actual management is or not quite sustainable. This issue should be clarified. The organization of the manuscript is improvable: Delete the title of Section 3 (Results) and add the text to former Section 4 Section 4 must be inserted before Section 2, and the generalities about this species only are linked to botanic issues. Maybe some other generalities regarding this species could be included here (cultural, etc.)….or change the title From former section 5 and beyond, it must be more evident that they are the results of the survey and interviews. These Sections could be renamed and fitted to the four lines included in the Abstract Section 7 should be joined with Table 1. Supplementary File S6 maybe must be included in the manuscript. In my view, it contains key issues regarding the management of this species. It sounds strange the title of Section 9 and a similar title for Section 10.1. Methods: Do the authors perform searches in Scopus and WOS using Spanish terms?

Minor comments:

226: put “Ñuu savi” in italics

Author Response

REVIEWER 3

This study addresses a study regarding the current status and the sustainable management of a palm species (Brahea dulcis) in Mexico. The aims and objectives of the paper fit in the editorial policy of this journal. I think the authors have made a worthy effort, and the manuscript is very promising. However, despite its evident strengths, my overall recommendation is that the manuscript should be not accepted now in its present form. My comments are listed as follows:

REVIEWER 3

The authors should clarify the term “survey” in the abstract.

AUTHORS

“Survey” was deleted

REVIEWER 3

 The term “sustainable management” appears in the title, and, profusely, in the last Section (“Further research”). For the reader, it is not entirely clear if the traditional and/or the actual management is or not quite sustainable. This issue should be clarified.

AUTHORS:

In the document we acknowledge that palm harvesting has proved to be sustainable (Lines 527-531, and 541-543), nevertheless to reach sustainability of the entire system (cultural, geographical and ecological dimensions), other issues must be considered as we discussed in section 4 (Lines 543-547; 559-563; 595-603; 607-621; 623-635)

REVIEWER 3

The organization of the manuscript is improvable: Delete the title of Section 3 (Results) and add the text to former Section 4 .

AUTHORS

It has been done.

REVIEWER 3

Section 4 must be inserted before Section 2, and the generalities about this species only are linked to botanic issues. Maybe some other generalities regarding this species could be included here (cultural, etc.)….or change the title

AUTHORS

Re-organization of the document was performed, as sections from 4-9 corresponded to results of the study, and as reviewer #3 mentioned this was not clear. See lines: 143 and 171.

REVIEWER 3

From former section 5 and beyond, it must be more evident that they are the results of the survey and interviews. These Sections could be renamed and fitted to the four lines included in the Abstract

AUTHORS

These suggestions were considered and included in the sections suggested. Lines: 143-507

REVIEWER 3

Section 7 should be joined with Table 1. Line: 222

AUTHORS

It has been done.

REVIEWER 3

Supplementary File S6 maybe must be included in the manuscript. In my view, it contains key issues regarding the management of this species.

AUTHORS

It has been done. Line 444

REVIEWER 3

It sounds strange the title of Section 9 and a similar title for Section 10.1.

AUTHORS

It was modified. See line 406

REVIEWER 3

Methods: Do the authors perform searches in Scopus and WOS using Spanish terms?

AUTHORS

This is correct.

REVIEWER 3

Minor comments:

226: put “Ñuu savi” in italics

AUTHORS:

It has been done.

This manuscript is a resubmission of an earlier submission. The following is a list of the peer review reports and author responses from that submission.

Round 1

Reviewer 1 Report

The authors present in this article an interesting experience of the cross-disciplinary study of ecological, cultural, and economic aspects of using of Brahea dulcis palms in subsistence of rural communities in Mexico. The materials of the study are based on an analysis of bibliographic sources as well as on the field researches of the authors. I recommend, if it's possible, to present more detailed description of this field research experience include brief description of the methods of collection of the ethnographic materials.

The results of the study have undoubted novelty and can by applied for solving of the modern problems of sustainability management based on natural adaptations of palms and involve planning the land where it is distributed in order to warrant its availability for local rural communities.

Author Response

Please see Attachment

Information relating methods of ethnographic survey and backgrounds of the authors is provided.

Reviewer 2 Report

The paper presents a thorough literature review of human use of the Soyate palm and presents the case that it is the most significant palm species used in central Mexico. The methods focus on a bibliographic review of the species which appears rigorous and thorough. The methods section also points out that the authors hold considerable knowledge and personal experience on the use of this species.

Overall the paper is well written, however there is a need for a thorough grammar check particularly in the latter half of the paper. For example 750 For what has been reported on management and use of palms in Mexico, B. dulcis is the native species with greater management intensity in the country.

One issue appears to be that the authors are summarising key findings from the literature as well as adding their own considerable personal knowledge as new 'unpublished data' in the results and discussion. Perhaps it would be better if this personal knowledge was better defined in the methods section (ie outlining some of the field experience of the authors that generated data) and if there was clearer attribution of this data in the results section (e.g. author pers. comm.).

The paper is quite long and some sections could be shortened by further summarising some results and citing literature for further reading.

Author Response

Reviewer #2

“...it would be better if this personal knowledge was better defined in the methods section (ie outlining some of the field experience of the authors that generated data) and if there was clearer attribution of this data in the results section (e.g. author pers. Comm.)”

Authors

Information relating methods of ethnographic survey and backgrounds of the authors is provided.

Reviewer # 2

The paper is quite long and some sections could be shortened by further summarising some results and citing literature for further reading”

Authors:

The table with information regarding palm species used and categories of use was removed from main text and provided as Supplementary File S2. Table with results of the bibliographic query was removed from main text and provided as Supplementary File S3. The section of “Post-harvest treatment for weaving” was also removed from main text and integrated with images relating palm weave objects in Supplementary File S5. The revised main text was reduced 5 pages.

Reviewer 3 Report

This topic is interesting, and has potential to add to the scientific literature, but it lacks scientific rigour needed to make is of significance to publish. It would be an adequate monograph on the species based on extensive experience of the authors. It fails to present a testable hypothesis. It fails to make adequate argument in support of the model. The methods outline a review of the literature, but then in the results the authors speak of field observations. Basically it is a review of 22 articles, with interjections of the authors experiences. The abstract is a collection of thoughts with little organization to the text. The text is fraught with grammatical and syntax challenges taht need to be addressed to allow the reader to clearly understand what the authors are trying to convey.

Author Response

Reviewer #3

“The abstract is a collection of thoughts with little organization to the text”

Authors:

The Abstract was reviewed to improve the information, and the coherence of the main ideas and proposals.

Reviewer #3

“ The methods outline a review of the literature, but then in the results the authors speak of field observations”

Authors:

Information relating methods of ethnographic survey and backgrounds of the authors is provided.

Reviewer #3

It fails to present a testable hypothesis. It fails to make adequate argument in support of the model.”

Authors:

The main aim of our study was to determine the state of the art about management of B. dulcis. We appreciate the reviewer’s comment very much. However, it is not our intention testing an experimental hypothesis. Rather, we aimed at understand main important subjects relating this important plant resource, specifically those related to the spatial dimensions of management, frequently disregarded, and to propose some relant outcomes for its sustainable management.

Please see the atachment.

Round 2

Reviewer 3 Report

the authors have made changes to the manuscript, but much more work needs to be done. There are significant grammatical challenges that detract from the readability and require major editing. There does not seem to be a general theme of the manuscript, but a collection of thoughts that are not woven together to present much science-based knowledge. Section 9 need considerable reorganization, as it does not add to knowledge about 'management' of the species. As example, section 9.7 'grounding' presents two sentences which do not provide the reader with much information about management. Section 10 is divided into ecological and spatial implications, neither of which provides implications into the management of the species. Section 11 (Discussion and Concluding remarks) then speaks to 'traditional management and its implications. The entire manuscript seems to be "organized" to throw a great deal of information at the reader in hopes that the reader will glean some useful knowledge. The very first reference is incomplete.

Author Response

Reviewer #3

Response

“The authors have made changes to the manuscript, but much more work needs to be done”

The present version of the document contains the recommendations of four different reviewers and those from the editorial section of the Journal. In addition, the support from IMPI (Editorial service of the journal) and the Academic Writing Office of UNAM has also made improvements. Authors are thankful for the significant improvement and we hope that the current version fulfills the Editors, Reviewers and Journal standards to be accepted for publication.  Revisions were
clearly highlighted, using the "Track Changes" function in
Microsoft Word, so that changes are easily visible to the editors and
reviewers.

“There are significant grammatical challenges that detract from the readability and require major editing”

Final English editing was conducted under the guidance of the service provided by the present journal support system and the Academic Writing Office (Susana Kolb) of UNAM and Editorial Service (IMPI). 

“Section 9 need considerable reorganization, as it does not add to knowledge about 'management' of the species. As example, section 9.7 'grounding' presents two sentences which do not provide the reader with much information about management. Section 10 is divided into ecological and spatial implications, neither of which provides implications into the management of the species”

 “The entire manuscript seems to be "organized" to throw a great deal of information at the reader in hopes that the reader will glean some useful knowledge”.

The hole section of “Brahea dulcis management” was improved, providing more information relating traditional management practices. The section of “Implications of management” informs about ecological and spatial implications of traditional management practices, leaving the implications for sustainability on section 11(Discussion).   Concluding sections were improve in order to provide a clearer and stronger argument relating implication of management for sustainability.

The present version of the document contains the recommendations of four different reviewers and those from the editorial section of the Journal. Authors are thankful for the significant improvement and we hope that the current version fulfills the Editors, Reviewers and Journal standards to be accepted for publication. 

The first reference is incomplete

It was corrected

Round 3

Reviewer 3 Report

I have reviewed this manuscript three times. Each time it is only moderately changed. At this point, it should be well polished, but I still find significant editorial challenges that imply that these will not be addressed. The reviewer should not be making recommendations on grammar and text. But, I find myself editing the piece. As example; in the first sentence of the abstract the use of 'of livelihood of several communities' should read for the livelihoods of communities worldwide. on line 24 'people life sustainability' makes no sense. I find it inappropriate to refer to the authors' positions, as you do in lines 142-150. The reader should assume that the authors are in positions of authority. The materials presented in lines 163-176 do not appear to be relevant for this study. The first sentence in section 3 [results] is not a sentence. Figure 2 appears to misrepresent the two phenotypes. Figure 2a is a low type, while Figure 2b is a tall type. But the text suggests otherwise. Table 3 has lots of parts presented, but it is unclear what 'domestic' use means. And most are of that type.

In a previous review, it was recommended that major reorganization be done. Clearly this is still needed. Section 5 deals with cultural importance. And section 8 deals with cultural importance. These sections should be combined.

Much of the discussion on 'management', section 9 is superfluous and really does not deal with management

In section 10.1 [ecological implications] does not really discuss the ecological implications of management. As example the discussion on line 534 is not ecological. In the paragraph starting on line 523 the authors make statements that should be supported by citations. In section 10.2 Management implications, the authors discuss ecological implications. This should be presented in the previous section.

The first sentence of section n11.1 seems to make little or no sense. And there are other examples where sentences make little sense.

The  figure 5 (model) is very elaborate, but the authors only make passing references to it. Something this significant should be thoroughly presented and discussed.

You may want to consider another journal, such as Economic Botany